# The Impact of Resilience on Post-Traumatic Growth among Nurses in COVID-19-Designated Hospitals: The Mediating Effect of Meaning in Life

**DOI:** 10.3390/healthcare11212895

**Published:** 2023-11-03

**Authors:** Suk-Jung Han, Young-Ran Yeun, Hyunseung Roh

**Affiliations:** 1College of Nursing, Sahmyook University, Seoul 01795, Republic of Korea; hansj@syu.ac.kr; 2College of Nursing, Kangwon National University, Samcheok 25649, Republic of Korea; 3Department of Public Health, Sahmyook University, Seoul 01795, Republic of Korea; rohhs.class@gmail.com

**Keywords:** COVID-19, resilience, post-traumatic growth, meaning in life

## Abstract

This study aimed to confirm the relationship between resilience, meaning in life, and post-traumatic growth (PTG) among nurses during the COVID-19 pandemic. In particular, the mediating effect of meaning in life on the relationship between resilience and PTG was examined. A cross-sectional descriptive research design was used. The participants were 220 nurses at COVID-19-designated hospitals in Seoul. Descriptive statistics, a *t*-test, ANOVA, Pearson’s correlation coefficient analysis, hierarchical regression, and SPSS PROCESS macro (Model 4) were used for data analyses. As a result of the study, resilience and meaning in life each had a significant positive effect on PTG. In addition, the indirect effect of meaning in life was significant, suggesting that meaning in life partially mediated the relationship between resilience and PTG. The results indicate that nurses’ resilience directly contributes to the improvement of PTG, and this relationship is further strengthened indirectly through the presence of meaning in life. Therefore, it is suggested that strategies should be developed to enhance resilience and promote a sense of meaning in the profession in order to support the mental health and foster growth among nurses.

## 1. Introduction

During the coronavirus disease 2019 (COVID-19) pandemic caused by the SARS-CoV-2 virus, the catastrophic situation of an infectious disease outbreak necessitated an increased role for healthcare professionals. Hospital nurses hold a critical position in the healthcare system because they constitute more than the majority of the healthcare workforce [1]. As frontline healthcare professionals who are in close contact and spend most of their time with infected patients, nurses faced greater challenges than other healthcare professionals, including physicians, during the global pandemic [2]. In Wuhan, China, in early 2020, the proportions of healthcare workers infected with COVID-19 were 52.1% among nurses and 33.6% among doctors [3].

In South Korea, nurses who cared for COVID-19 patients reported negative emotions such as anxiety, fear of infection and contagion, and emotional exhaustion; however, they simultaneously felt recognized by those around them, and their professional identity matured [4]. Conversely, individuals who have experienced trauma may experience distress or maladjustment, but they may also appreciate life more than before the traumatic event, discover new possibilities and meanings in their lives, and reset their life goals. Additionally, they may value intimate relationships more, perceive themselves as stronger as they work through their pain, and experience an increase in spiritual interest and depth [5].

Post-traumatic growth (PTG) refers to positive psychological changes that occur as a result of coping with or struggling with stressful events [6]. Tedeschi and Calhoun [5] found that some individuals who experienced a traumatic event experience psychological growth after the traumatic event that exceeds their pre-traumatic level of functioning owing to their internal resources and social support, which is the result of their cognitive process of adapting to a changed life after the traumatic event. Several factors reportedly influence PTG, including resilience and meaning in life [7,8].

Resilience is the ability to transform adversity into a mature experience [9], grow mentally when faced with difficulties, and adapt to the environment [10,11,12]. It refers to the ability to effectively recover from stressful situations when faced with adversity [13]. Resilience can assist individuals in coping with stressors associated with increased mental health problems such as worry [14]. Consequently, promoting resilience can improve mental health and positive functioning [15]. A previous study on nurses found that higher levels of resilience were associated with more positive stress coping and greater adaptability in stressful situations [16]. Resilience also positively predicted PTG [17] and appears to play an important role in the development of PTG [18].

Meaning in life has been defined as coherence in life, goal orientation or purposefulness, intention to act, and the reason for acting [19,20]. Meaning in life assists individuals in coping with suffering when faced with adversity or misfortune by giving it meaning. Suffering stimulates the need for meaning [21]. Meaning in life is highly related to mental health [22] and has been reported to have positive effects on adaptation and coping; that is, individuals with meaning in life experience greater happiness, satisfaction, and positive emotions; meaning in life promotes adaptive coping, especially by moderating or mediating the relationship between trauma or stress and mental health [23]; and meaning in life has been reported to act as a buffer resource to experience hope or PTG, despite experiencing adversity or stress [24,25]. Previous studies have reported that meaning in life positively affects PTG [7,26].

Kim and Shin [7] found that individuals with self-resilience dispositions experience negative stressful events, such as trauma, and then overcome them and engage in a process of post-traumatic growth. They found that this process is mediated by searching for and discovering meaning in life. Song and Lee [25] found that meaning in life fully mediated the relationship between mothers’ hope and post-traumatic growth in children with cancer. Kim and Kim [27] found an indirect mediating effect of meaning in life on the relationship between social support and post-traumatic growth. There has not been much research on the mediating effect of meaning in life. Kim and Cho [28] found that emotion recognition clarity had a static effect on post-traumatic growth, mediated by meaning in life. As such, there are several studies that have shown that many factors affect post-traumatic growth, both directly and indirectly through quality in life parameters.

This study aimed to examine the mediating effect of meaning in life on the relationship between resilience and post-traumatic growth among nurses who worked in COVID-19-dedicated hospitals during the pandemic.

## 2. Materials and Methods

### 2.1. Study Design

This study was designed as a cross-sectional study to determine the mediating effect of meaning in life on the relationship between resilience and PTG among hospital nurses during the pandemic.

### 2.2. Study Participants

The participants in this study were nurses working in three COVID-19-designated public hospitals in Seoul, South Korea. The specific inclusion criteria were the following: (1) nurses with at least 1 month of work experience wearing personal protective equipment (PPE) during the pandemic up to the time of study entry; and (2) nurses with at least 6 months of clinical experience.

The appropriate sample size required for data analysis was calculated for multiple linear regression analysis using the G*Power 3.1.9.7 program. For effect size = 0.15 (median), significance level (α) = 0.05, power (1 − β) = 0.95, and considering 13 predictors, the required minimum number of subjects was 189, and a total of 220 responses were used in the analysis.

This study was conducted with the permission and cooperation of the nursing departments of three dedicated COVID-19 public hospitals in Seoul: one major public hospital in Seoul and two branch hospitals. The researcher protocol required the researchers to detail the purpose and methodology of the study to the nursing departments of the three hospitals, and then the head nurse explained the same to the nurses in each department. Data were collected from May to the end of July 2021 from nurses caring for COVID-19 patients in dedicated COVID-19 hospitals.

### 2.3. Ethical Consideration

This study was approved by the Institutional Review Board of Seoul National University Medical Center, Seoul, Korea (SEOUL 2021-04-006). A study information sheet, consent form, and questionnaire were provided to participants individually. The information sheet included information about the purpose of the study, data collection methods, duration of the study, researcher’s name and contact information, consent to participate in the study, right to refuse participation, and personal confidentiality. In particular, it was explained that all data would be anonymized and would not be used for any purpose other than the study, and that participants could withdraw from the study at any time during the study without penalty, even after stopping or completing the survey.

Written consent was obtained before answering the questionnaire, and the questionnaire was marked with symbols to protect the privacy of the research participants. For completing the questionnaire, participants were offered a drink voucher worth 10,000 KRW.

### 2.4. Measures

#### 2.4.1. Resilience

Resilience refers to the psychological and social characteristics of individuals that enable them to cope, adapt, and grow when faced with serious life challenges such as hardship or adversity [29].

In this study, resilience was measured with the Korean version of the Connor–Davidson Resilience Scale (K-CD-RISC), an instrument developed by Connor and Davidson [29] and standardized into Korean by Baek et al. [30]. The scale consists of 25 items and 5 sub-factors: 8 for hardiness, 8 for persistence, 4 for optimism, 2 for support, and 2 for spirituality. Each item is rated on a 5-point Likert scale ranging from 0 (not at all) to 4 (very much so), with higher scores indicating greater resilience. The reliability of the instrument was Cronbach’s α = 0.93 in the study by Back et al. [30] and 0.952 in this study.

#### 2.4.2. Meaning in Life

Meaning in life refers to the motivation to seek meaning in life, and it is divided into two dimensions: ‘meaning presence’ and ‘meaning search’. Meaning presence is the subjective feeling that one’s life is meaningful, and meaning search is the orientation to find meaning in one’s life [31].

In this study, meaning in life was measured with the Meaning in Life Questionnaire (MLQ), an instrument originally developed by Steger et al. [32] and translated into Korean and validated by Won et al. [33]. The MLQ consists of 10 items and 2 sub-factors: 5 for the presence of meaning in life and 5 for the search of meaning in life. Each item is rated on a 7-point Likert scale ranging from 1 (never) to 7 (always), with higher scores indicating a more positive presence and search of meaning in life. In Won et al.’s [33] study, the subscales indicated a Cronbach’s alpha of 0.82 and 0.87, respectively, and in this study, the Cronbach’s alpha was 0.918 for the entire tool, and 0.843 and 0.926 for the subscales.

#### 2.4.3. Post-Traumatic Growth

PTG is a positive psychological change that goes beyond overcoming trauma and refers to qualitative changes in personal functioning and adaptation that go beyond the ability to resist trauma or pre-trauma levels [8].

In this study, PTG was measured with the Korean version of the Post-traumatic Growth Inventory-Expanded (PTGI-X), an instrument developed by Tedeschi et al. [6], translated into Korean by Song et al. [34], and standardized by Kim et al. [35].

The scale consists of 25 items and 4 sub-factors: 8 for change in self-perception, 5 for an increase in interpersonal depth, 7 for an increase in spiritual and existential depth, and 5 for the discovery of new possibilities. Each item is rated on a 6-point Likert scale ranging from 0 (not at all) to 5 (very much so), with higher scores indicating a more positive post-traumatic change. At the time of development, Cronbach’s alpha was 0.97 in a study by Kim et al. [35] and 0.969 in this study.

### 2.5. Statistical Analysis

The collected data were analyzed using SPSS (version 25.0) and the PROCESS macro (version 4.0) using Model 4. Frequency analysis and descriptive statistical analysis were conducted to identify the general characteristics of the subjects and the main study variables. The differences in ‘post-traumatic growth’ according to general characteristics were subjected to a *t*-test, ANOVA, and Scheffé analysis for post-hoc testing. Partial correlation analysis was conducted to examine the correlation between the study variables while controlling for age and experience, which showed significant differences in post-traumatic growth among general characteristics. To determine the mediating effect of the meaning in life variable on the effect of resilience on post-traumatic growth, a regression analysis was conducted using Hayes’ [36] PROCESS macro model 4. Bootstrapping was performed 5000 times, and the bias-corrected bootstrap confidence interval was set at 95% to calculate the direct and indirect effect sizes, and the mediating effect was considered statistically significant if the confidence interval did not include ‘0’.

## 3. Results

### 3.1. General Characteristics of Subjects and Differences in PTG

The general characteristics of the participants are shown in Table 1. The average age of the 220 participants was 32.1 ± 7.3 years, 48.2% (n = 106) were in their 20s, and 92.7% (n = 204) were female. In addition, 78.2% (n = 172) had a bachelor’s degree, 32.3% (n = 71) were religious, 69.5% (n = 153) were single, and 29.5% (n = 65) lived alone. In addition, 15.5% (n = 34) had experienced self-isolation due to COVID-19, 86.8% (n = 191) had cared for someone with COVID-19, and 39.1% (n = 86) had worked in personal protective equipment for 151–250 days. The mean total work experience was 7.9 ± 6.9 months (Table 1).

When comparing the PTG scores according to participants’ general characteristics, we found that higher age groups tended to have higher PTG scores, with a significant difference between those aged 20 and under and those aged 50 and over (F = 5.16, *p* = 0.002). Religious subjects tended to have significantly higher PTG scores than non-religious subjects (t = −3.79, *p* < 0.001), and married subjects tended to have significantly higher PTG scores than single subjects (t = −5.04, *p* < 0.001).

There were no significant differences in the PTG scores by gender, education level, cohabitation status, experience with self-quarantine, experience with caring for patients with COVID-19, duration of work wearing PPE, and total career.

### 3.2. Correlations between Resilience, Meaning of Life, and PTG

Participants’ resilience averaged 56.97 ± 15.32 out of 100, meaning in life averaged 46.04 ± 10.14 out of 70, and PTG averaged 59.73 ± 24.83 out of 125. As a result of examining the skewness and kurtosis values of the study variables, all were distributed within ±1, satisfying the assumption of normal distribution.

The correlations between resilience, meaning in life, and PTG were analyzed and it was found that PTG was positively correlated with resilience (r = 0.49, *p* < 0.001) and meaning in life (r = 0.54, *p* < 0.001). Therefore, the higher the resilience and meaning in life, the higher the PTG (Table 2).

### 3.3. The Mediating Effect of Meaning in Life on the Relationship between Resilience and PTG

To test the mediating effect of meaning in life on the relationship between resilience and PTG, significance tests were conducted for each pathway after controlling for age, religion, marital status, and years of work experience, which were expected to influence PTG among the general characteristics (Figure 1).

The results of the direct effect test indicated that resilience was positively related to meaning in life (β = 0.59, *p* < 0.001). When analyzed with resilience as the predictor and meaning in life as the dependent variable, higher resilience was associated with higher meaning in life. When analyzing resilience as a predictor and PTG as a dependent variable, resilience had a significant direct effect on PTG (β = 0.44, *p* < 0.001). When analyzed using meaning in life as a predictor and PTG as the dependent variable, meaning in life had a static effect on PTG (β = 0.34, *p* < 0.001).

The mediating effect of resilience on PTG through meaning in life was significant (B = 0.02, 95% bootstrap CI [0.12, 0.29]) because the 95% bootstrap confidence interval did not include zero. The indirect effect of meaning in life was also significant, suggesting that meaning in life had a partial mediating effect on the relationship between resilience and PTG (Figure 1, Table 3).

## 4. Discussion

This study explored the mediating effect of meaning in life on the relationship between nurses’ resilience and PTG to provide a basis for establishing strategies to increase PTG to promote nurses’ mental health.

The results of this study showed that resilience not only had a direct effect on improving post-traumatic growth, but also had an indirect effect through meaning in life. In their review of PTG, Gower et al. [37] emphasized that uncovering the mechanisms of PTG, a dynamic process of change, is important for interventions in high-risk populations. In other words, to better support nurses’ PTG, it is important to understand how individuals adapt and cope with adversity, particularly those facing serious challenges or risks, such as COVID-19. By studying these mechanisms, interventions and strategies can be developed to promote PTG among clinical nurses and help them overcome difficult situations. This study is unique in that it extends the research that has been limited to examining the relationship between resilience and PTG [38,39,40,41] by showing a model in which resilience makes a double contribution to PTG through meaning in life.

A discussion based on these results is as follows. First, higher resilience is associated with greater positive changes after a traumatic event. A previous study that examined the relationship between emergency department nurses’ work environment, relationship with the charge nurse, resilience, and PTG [42] found that resilience was the most important variable for PTG. Luo et al. [43] reported that, in medical students, stressful experiences, such as COVID-19, promoted resilience, which provides protection against stress and, in turn, stimulates PTG. Furthermore, Lyu et al. [44] found that healthcare workers with high levels of resilience demonstrated an increased use of positive coping strategies, which facilitated cognitive reappraisal and, in turn, promoted PTG. Thus, if nurses are resilient in the face of a stressful situation and able to overcome and adapt to it, they will experience a positive psychological growth that exceeds pre-traumatic levels rather than simply returning to pre-traumatic levels. Therefore, PTG is dependent on the ability to transform stressful situations into growth experiences based on toughness, perseverance, and optimism [45]. Consequently, interventions and training to increase nurses’ resilience should continue to be provided. Resilience is not innate but can be enhanced through training and education, and interventions to increase resilience need to be developed.

Second, higher resilience was associated with higher meaning in life. A previous study exploring the relationship between resilience and meaning in life among 1422 Spanish healthcare workers [46] supported this study’s findings that resilience predicts meaning in life. Lasota and Mróz [47] also found a positive relationship between resilience and meaning in life during the COVID-19 pandemic. Conversely, Yıldırım et al. [48] found that individuals with a high meaning in life were more resilient when faced with setbacks or trauma. Tang et al. [49] reported that a meaningful life could be a protective factor that enhances resilience in nurses. Therefore, resilience and meaning in life were positively correlated in both directions.

Finally, the mediating effect of meaning in life on the relationship between resilience and PTG was identified. The effects of resilience on PTG were predicted after controlling for other positive demographic variables. Overall, the findings suggest that nurses with greater resilience experience more PTG and that the relationship between resilience and PTG may be mediated by meaning in life. Several studies have noted that people who are exposed to stressful and potentially traumatic events experience both positive and negative post-traumatic changes [50,51]. For example, people who become more independent after a traumatic event may feel more vulnerable than they thought they were. People who learned that others are caring and helpful may also have learned how cold and selfish some people can be. However, it can be inferred that individuals with high resilience have a clear sense of meaning in life, which has a strong positive impact on their post-traumatic recovery by enabling them to cope proactively rather than avoiding difficulties. These findings are important because they address the important issue of post-traumatic growth in people who often perceive trauma as negative due to the experience of adverse events, but who face significant risks such as COVID-19, and suggest the role of resilience models in reflecting on resilience and post-traumatic growth research for individuals and the wider community. They are also significant because they integrate the fragmented relationships between variables in previous studies and provide a clearer picture of these relationships. The mediating effect of meaning in life on the relationship between resilience and PTG is particularly significant given that individuals who have experienced a traumatic event experience significant confusion regarding the meaning of their lives [52,53,54].

Research findings on the relationship between nurses’ resilience, meaning in life, and post-traumatic growth can be used to guide various strategies and interventions to support nurses in their personal and professional development. Health care institutions and organizations can incorporate elements focused on resilience and finding meaning of life into their nursing education programs. Nurses can receive resilience training programs to help develop and strengthen their resilience skills. Additionally, healthcare organizations can implement meaning-focused interventions to help nurses find deeper purpose and meaning in their work. In other words, by strengthening nurses’ resilience and promoting meaning in their profession, healthcare organizations can help nurses deal with the challenges they face and provide better care to patients. In addition, the results of this study can be used for measurement and evaluation by organizational management. Healthcare organizations can incorporate the measurement of resilience and the meaning of life in their assessment of nursing staff’s well-being and job satisfaction. These data can inform targeted interventions and provide support for those who may be struggling in these areas.

The limitations of this study and recommendations are as follows. First, this study assumed only meaning in life as a mediator, and no single mediator can explain the relationship between resilience and PTG. Further studies are required to explore the variables that may serve as mediators in promoting nurses’ PTG. Second, this study did not identify group differences in the various sub-factors within the three variables; therefore, further studies are required to better understand the differences between the subtypes of the variables. Third, because this study is cross-sectional, there may be limitations in explaining the temporal relationship and causal relationship between each factor. Therefore, future research should apply systematic research methods to reveal causal relationships.

## 5. Conclusions

The results of this study provide important insights for the development of strategies aimed at enhancing the mental health and PTG of nurses. According to the research findings, nurses’ resilience directly contributes to the improvement of PTG, and this relationship is further strengthened indirectly through the presence of meaning in life. This information is particularly relevant for individuals interested in fostering psychological adaptation and PTG among various professional groups, including nurses who work in high-risk situations. Therefore, this study underscores the significance of both resilience and the presence of meaning in life and is expected to contribute to the development of strategies aimed at supporting the mental health of nurses and promoting personal growth.

## Figures and Tables

**Figure 1 healthcare-11-02895-f001:**
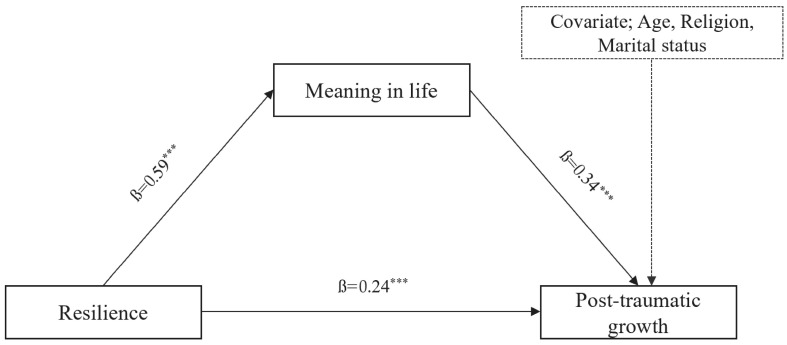
Mediating effect model of resilience on post-traumatic growth through meaning in life. Adjusted for age, religion, and marital status. All path coefficients are standardized regression weights. Total effect = 0.44; direct effect = 0.24; indirect effect = 0.20 (95% BC bootstrap CI = 0.12–0.29), *** *p* < 0.001.

**Table 1 healthcare-11-02895-t001:** Differences in post-traumatic growth according to participants’ characteristics (N = 220).

Characteristics	Categories	N (%)		Post-Traumatic Growth			
				M ± SD	t/F	*p*	Scheffé
Age (year) M ± SD = 32.1 ± 7.3	≤29	106	(48.2)	53.83 ± 24.33 ^a^	5.16	0.002	a < b
	30–39	78	(35.5)	62.91 ± 26.09 ^ab^			
	40–49	30	(13.6)	68.46 ± 19.51 ^ab^			
	≥50	6	(2.7)	79.16 ± 8.37 ^b^			
Gender	female	204	(92.7)	59.60 ± 24.73	0.26	0.794	
	male	16	(7.3)	61.43 ± 26.80			
Educational background	associate degree	41	(18.6)	55.87 ± 25.94	2.42	0.091	
	bachelor’s degree	172	(78.2)	59.91 ± 24.67			
	master’s degree	7	(3.2)	78.00 ± 13.26			
Religion	have	71	(32.3)	68.66 ± 24.23	−3.79	<0.001	
	do not have	149	(67.7)	55.48 ± 24.04			
Marital status	married	67	(30.5)	71.08 ± 20.73	−5.04	<0.001	
	not married	153	(69.5)	54.76 ± 24.90			
Cohabitation	live alone	65	(29.5)	56.56 ± 26.55	−1.22	0.221	
	not live alone	155	(70.5)	61.06 ± 24.04			
Self-quarantine experience	yes	34	(15.5)	64.14 ± 23.58	1.17	0.246	
	no	186	(84.5)	58.93 ± 25.03			
Infected patient nursing management	yes	191	(86.8)	59.39 ± 24.96	−0.51	0.605	
	no	29	(13.2)	61.96 ± 24.24			
Duration of work wearing PPE (day)	≤50	21	(9.5)	68.95 ± 21.87	1.53	0.207	
	51–150	74	(33.6)	56.59 ± 25.62			
	151–250	86	(39.1)	59.02 ± 24.12			
	>251	39	(17.7)	62.30 ± 25.79			
Total career (year) M ± SD = 7.9 ± 6.9	≤1	18	(8.2)	61.11 ± 26.57	2.26	0.50	
	1~3	47	(21.4)	56.31 ± 24.68			
	3~5	38	(17.3)	50.78 ± 27.80			
	5~10	55	(25.0)	60.96 ± 22.99			
	10~20	51	(23.2)	65.17 ± 23.03			
	≥20	11	(5.0)	71.63 ± 21.36			

M = Mean; SD = Standard deviation. PPE = personal protective equipment. ab, a, b means the result of post-hoc test of Scheffe.

**Table 2 healthcare-11-02895-t002:** Correlations among resilience, meaning in life, and post-traumatic growth (n = 220).

Variables	RES	MIL	PTG	Min	Max	M ± SD	Skewness	Kurtosis
		r (*p*)						
RES	1			21.00	100.00	56.97 ± 15.32	0.02	−0.35
MIL	0.60 (<0.001)	1		13.00	70.00	46.04 ± 10.14	−0.33	0.43
PTG	0.49 (<0.001)	0.54 (<0.001)	1	5.00	114.00	59.73 ± 24.83	−0.29	−0.62

RES = Resilience; MIL = Meaning in Life; PTG = Post-traumatic Growth; M ± SD = Mean ± Standard deviation.

**Table 3 healthcare-11-02895-t003:** Mediating effects of meaning in life on the relationship between resilience and post-traumatic growth (n = 220).

Path	B	SE	β	t (*p*)	Adj.R^2^	F (*p*)	Indirect Effect	
							Boot LLCI	Boot ULCI
RES→MIL	0.39	0.03	0.59	10.86 (<0.001)	0.387	22.43 (<0.001)		
MIL→PTG	0.84	0.16	0.34	5.08 (<0.001)	0.404	20.54 (<0.001)		
RES→PTG	0.39	0.11	0.24	3.58 (<0.001)				
RES→MIL→PTG	0.20	0.04					0.12	0.29

RES = Resilience; MIL = Meaning in Life; PTG = Post-traumatic Growth; M = mean; SD = standard deviation; β = standardized estimates; Adj.R^2^ = adjusted R^2^; LLCI = lower level of the 95% confidence interval; ULCI = upper level of the 95% confidence interval.

## Data Availability

Please contact the corresponding author for data availability.

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
