# Peer review of "The Impact of Resilience on Post-Traumatic Growth among Nurses in COVID-19-Designated Hospitals: The Mediating Effect of Meaning in Life"

_healthcare, 2023, doi:10.3390/healthcare11212895_

Round 1
Reviewer 1 Report
Comments and Suggestions for Authors
healthcare-2627845. The Impact of Resilience on Post-traumatic Growth among 2 Nurses in a COVID-19 Dedicated Hospital: the Mediating Ef-3 fect of Meaning in Life
Thank you for the opportunity to review this study, which was aimed to: investigate how meaning in life, which focuses on the process of meaning-seeking and discovery, affects growth through self-resilience, a personal characteristic that contributes to PTG, and personal efforts to cope with trauma.
The topic is interesting and the results are valuable. However, some concerns preclude publication as presented.
MAJOR COMMENTS
INTRODUCTION
The reader would benefit if definitions are clarified, according to the instruments selected to perform the study; while avoidance of mixed concepts is desirable
The aims require clear statements on the study variables, including co-factors, without introducing variables that were not assessed in the study protocol.
METHODS
Better description of the circumstances in which the participants gave response to the questionnaires is required, including the setting and the history of vaccination.
Appropriate description of all the instruments is needed.
The statistical analysis subsection should include just the analysis performed with the appropriate consideration of the test assumptions (such as normal distribution and unequal N by subgroups) as well as the significance level, instead of a discussion of different methods. The first paragraph of this section suggests Normal distribution, while the last paragraph suggests that the distribution was different than Normal.
RESULTS
Repetitive descriptions are not helpful, but confusing.
The description of simple correlations is incomplete, the age and experience are missing.
It is recommended to describe the main results in the text, while full description is provided in tables, which may display results on the main variables by sex.
DISCUSSION
The reader would benefit if the authors discuss the full results of modelling, which shows double contribution of resilience to posttraumatic-growth.
It is highly suggested that other limitations are considered (including the design that preclude statements on causal relationships).
Conclusions should be supported by the findings; while recommendations could be provided in a separate paragraph at the end of the discussion.
MINOR COMMENTS
A revision of the language use (e.g. this study aimed instead of this study was aimed (L80) among others), and clarification of some sentences (e.g. 39.1% (86) had held a D-level (L169) for … , among some others) is required.
Some paragraphs have lost the Layout and some sentences are isolated.
Comments on the Quality of English LanguageThe manuscript requieres language revision.
Author Response
We thank you for your detailed and valuable comments. We have revised the paper to address the issues raised to the best of our ability and have marked the revised sections in red. We have also tabulated our responses to each comment. Please see the attachment for details.

Reviewer 2 Report
Comments and Suggestions for Authors
I have received for review a manuscript titled “The Impact of Resilience on Post-traumatic Growth among 2 Nurses in a COVID-19 Dedicated Hospital: the Mediating Effect of Meaning in Life”. It is well developed manuscript; however, the following points will help improve the quality further;
1. In the abstract section must not contain the significance values and beta values while reporting the results. The authors may write in descriptive form instead.
2. Single line implications of the results are required in the abstract section.
3. Single line originality description is
4. The paragraph written on Line Number 48, 49 is too short.
5. The paragraph written on Line Number 55, 56 is too short.
6. The introduction section along with the literature review section is not articulated. The authors must explain the need for the study, what the earlier studies have told about the relationships to examine. What is the significance of the topic in the global scenario, research gaps, contributions of the study, significance of the study and sequence of the study.
7. The methodology is deficient. The authors have missed out writing about the population of Nurses, the method used for sample selection, unit of analysis, sample size, how have they selected the sample size? Which technique they have employed? And so forth.
8. Data collection procedure from the Nurses is a missing element in the methodology section. How much time they have invested in gathering data? It is believed that Nursing is a hectic and job and a job that requires too much attention and commitment. How the authors have managed to get time from Nurses to participate in data collection for their study?
9. The authors have missed out writing the statements to check each variable in the methodology section.
10. The mediation results are missing. Must be presented in a table.
11. Discussion is too short.
12. Implications are missing.
13. Limitations and future directions are missing.
14. I have not checked the plagiarism of this document, it stands the responsibility of the authors to examine it before submission.
Decision Major Changes
Comments on the Quality of English Language
I have received for review a manuscript titled “The Impact of Resilience on Post-traumatic Growth among 2 Nurses in a COVID-19 Dedicated Hospital: the Mediating Effect of Meaning in Life”. It is well developed manuscript; however, the following points will help improve the quality further;
1. In the abstract section must not contain the significance values and beta values while reporting the results. The authors may write in descriptive form instead.
2. Single line implications of the results are required in the abstract section.
3. Single line originality description is
4. The paragraph written on Line Number 48, 49 is too short.
5. The paragraph written on Line Number 55, 56 is too short.
6. The introduction section along with the literature review section is not articulated. The authors must explain the need for the study, what the earlier studies have told about the relationships to examine. What is the significance of the topic in the global scenario, research gaps, contributions of the study, significance of the study and sequence of the study.
7. The methodology is deficient. The authors have missed out writing about the population of Nurses, the method used for sample selection, unit of analysis, sample size, how have they selected the sample size? Which technique they have employed? And so forth.
8. Data collection procedure from the Nurses is a missing element in the methodology section. How much time they have invested in gathering data? It is believed that Nursing is a hectic and job and a job that requires too much attention and commitment. How the authors have managed to get time from Nurses to participate in data collection for their study?
9. The authors have missed out writing the statements to check each variable in the methodology section.
10. The mediation results are missing. Must be presented in a table.
11. Discussion is too short.
12. Implications are missing.
13. Limitations and future directions are missing.
14. I have not checked the plagiarism of this document, it stands the responsibility of the authors to examine it before submission.
Decision Major Changes
Author Response

(The authors gave the same response as above.)

Round 2
Reviewer 1 Report
Comments and Suggestions for Authors
Improvement was evident in all the sections of the manuscript.
Comments on the Quality of English LanguageMinor typos to be revised
Author Response
Thank you for your comments to help us improve the manuscript. The sentence structure and typographical errors you mentioned have been changed and marked in red. Thank you once again for your careful attention and consideration.